The effects of growth rate and biomechanical loading on bone laminarity within the emu skeleton

Kuehn Amanda L. 1
Lee Andrew H. 2
Main Russell P. 3
Simons Erin L.R. esimon@midwestern.edu 4
1 Arizona College of Osteopathic Medicine, Midwestern University , Glendale , AZ , United States of America
2 Department of Anatomy, College of Graduate Studies, Arizona College of Osteopathic Medicine, College of Veterinary Medicine, Midwestern University , Glendale , AZ , United States of America
3 College of Veterinary Medicine, Purdue University , West Lafayette , IN , United States of America
4 Department of Anatomy, College of Graduate Studies, Arizona College of Osteopathic Medicine, Midwestern University , Glendale , AZ , United States of America
Hutchinson John
Electronic publication date: 2019 Sep 25
Publication date: 2019
Volume: 7
Electronic Location ID: e7616
Received 2018 May 8; Accepted 2019 Aug 5
Copyright: ©2019 Kuehn et al.
Copyright year: 2019
Copyright holder: Kuehn et al.
License: This is an open access article distributed under the terms of the Creative Commons Attribution License, which permits unrestricted use, distribution, reproduction and adaptation in any medium and for any purpose provided that it is properly attributed. For attribution, the original author(s), title, publication source (PeerJ) and either DOI or URL of the article must be cited.
License URL: https://creativecommons.org/licenses/by/4.0/

Keywords: Laminarity, Growth rate, Torsion, Shear strain, Bone, Emu

Funding: Midwestern University College of Health Sciences Biomedical Sciences Chapman Fund (Harvard University) This work was supported by Midwestern University College of Health Sciences Biomedical Sciences graduate funding to Amanda Kuehn. Long-term storage of the samples was supported by the Chapman Fund (Harvard University). The funders had no role in study design, data collection and analysis, decision to publish, or preparation of the manuscript.

==============================
The orientation of vascular canals in primary bone may reflect differences in growth rate and/or adaptation to biomechanical loads. Previous studies link specific canal orientations to bone growth rates, but results between different taxa are contradictory. Circumferential vascular canals (forming laminar bone) have been hypothesized to reflect either (or both) rapid growth rate or locomotion-induced torsional loading. Previous work on the hindlimb biomechanics in the emu shows that the femur and tibiotarsus experience large shear strains, likely resulting from torsional loads that increase through ontogeny. Here, we test how growth rate and biomechanical loading affect bone laminarity in wing and hindlimb elements from growing emu (2–60 wks). If laminar bone is an adaptation to torsion-induced shear strains, it should increase from juveniles to adults. Alternatively, if bone laminarity reflects rapid growth, as has been shown previously in emu, it should be abundant in fast-growing juveniles and decrease with age. Transverse mid-shaft histological sections from the limb bones (femur, tibiotarsus, humerus, ulna, and radius) were prepared and imaged. Growth rates were measured using fluorescent bone labels. Vascular canal orientation was quantified using laminarity index (proportion of circumferential canals). Principal components analysis was performed to convert highly correlated variables (i.e., mass, age, growth rate, and shear strain) into principal components. Random-intercept beta regression modeling determined which principal components best explained laminarity. The fastest growth rates were found in young individuals for all five skeletal elements. Maximum growth rate did not coincide with peak laminarity. Instead, in the femur and tibiotarsus, elevated laminarity is strongly correlated with adult features such as large size, old age, and modest growth rate. This result is contrary to predictions made based on a previous study of emu but is consistent with results observed in some other avian species (penguin, chicken). Shear strain in the caudal octant of the femur and tibiotarsus is positively correlated with laminarity but has a weaker effect on laminarity relative to mass, age, and growth rate. Laminarity in the wing elements is variable and does not correlate with ontogenetic factors (including mass, age, and growth rate). Its presence may relate to relaxed developmental canalization or a retained ancestral feature. In conclusion, ontogeny (including growth rate) is the dominant influence on vascular canal orientation at least in the hindlimb of the emu.

Introduction

Avian bone tissue is highly vascularized with a fibrolamellar structure, which allows for rapid growth by depositing randomly arranged spicules of woven bone initially, followed by in-filling of the cancellous spaces with centripetal lamellar bone, forming primary osteons (Francillon-Vieillot et al., 1990; de Ricqlès et al., 1991; Curry, 2002). Each primary osteon contains a central canal that houses blood vessels and nerves. These vascular canals vary in orientation and bones can be classified based on the predominant canal orientation. Laminar bone has a higher proportion of canals with circumferential orientation (parallel to the periosteal surface of the bone) relative to other orientations. Additional canal orientations include: radial, those orthogonal to the periosteal surface; longitudinal, those running parallel to the long axis of the bone; and oblique, all other orientations (de Ricqlès et al., 1991).

It has been hypothesized that differences in primary vascular canal orientation might be a reflection of growth rate, biomechanical loads, or phylogenetic relationships (Padian, 2013). Amprino (1947) first suggested that the organization of bone microstructure may be influenced by bone growth rate, such that woven bone is deposited during rapid growth and lamellar bone during slow growth (de Ricqlès et al., 1991). Further studies have investigated whether specific primary vascular canal orientations in fibrolamellar bone are also associated with slow or fast growth by directly comparing microstructure with bone growth rates measured through the use of injectable fluorochromes (Castanet et al., 2000; de Margerie, Cubo & Castanet, 2002; de Margerie et al., 2004). Rapidly growing hindlimb bones of ratites have been found to exhibit structure that is laminar and reticular (bone with numerous obliquely-oriented canals), whereas the more modest-growing wing elements of ratites exhibit reticular and longitudinal canal structure (Castanet et al., 2000). This suggests that laminar bone, in part, may reflect faster growth rates. This result was supported in a recent study of pigeon wing elements, which showed that peak laminarity (proportion of laminar bone) coincides roughly with the growth spurt in each element (Ourfalian, Ezell & Lee, 2016). However, work on mallard long bones showed no relationship between growth rate and predominant vascular canal orientation (de Margerie, Cubo & Castanet, 2002). Additionally, in the king penguin, radially-oriented canals dominated in the fastest growing sections, not circumferential canals (laminar bone) (de Margerie et al., 2004). Likewise, chickens selected for fast growth showed limb bones with predominantly radial canals (Williams et al., 2004; Pratt & Cooper, 2018).

Laminar bone has been hypothesized to better resist torsional loading. In laminar bone, the bone tissue is arranged in ‘sheets’ or ‘plates’ between layers of circumferential canals. Shear strain is thought to flow continuously within these ‘sheets’, and thus the concentrated stresses on the bone tissue surrounding the canals is reduced (de Margerie et al., 2004). Indeed, bone elements that are hypothesized to predominantly experience torsional loads have been found to exhibit laminar bone. Laminar bone is found to be most abundant in the humerus, ulna, and femur in a large sample of flighted bird species (de Margerie, 2002; de Margerie et al., 2005). In vivo strain gauge studies have shown that these elements experience predominantly torsional loads in at least some species: the humerus in the pigeon during flapping flight, the ulna in the turkey during wing flapping while on the ground, and the femur in the chicken and emu during terrestrial running, (Rubin & Lanyon, 1985; Biewener & Dial, 1995; Carrano & Biewener, 1999; Main & Biewener, 2007).

A limitation of previous studies of laminar bone is the indirect comparison of bone histology in one species with bone growth rates and/or in vivo strain gauge measures taken from different species. In this study, we present an analysis of laminar bone in a species in which bone growth rate and in vivo bone strain data were directly measured. The emu (Dromaius novaehollandiae, Order Struthioniformes, Family Dromaiidae) is a flightless bird endemic to Australia, but widely farmed in the US. The individuals included in this study comprise a growth series that were previously injected with fluorescent bone labels and surgically implanted with gauges to measure in vivo locomotor strains in the femur and tibiotarsus (Main & Biewener, 2007). Shear strains, produced by torsional loads about the long axes of the bones, were the predominant type of strain in the two bones and increased from juveniles to adults (Main & Biewener, 2007). Bone strains were not measured in the wing elements of these individuals. Presumably, shear strains are negligible in the wing elements because emus have extremely reduced wings, which have no known function other than occasionally being raised to aid thermoregulation (del Hoyo, Elliot & Sargatal, 1992; Maxwell & Larsson, 2007).

Therefore, if laminarity is an adaptation to torsion-induced shear strains, we predict that hindlimb bone laminarity will increase from juveniles to adults. Consistent with this hypothesis, we expect no trend in laminarity in the vestigial wing elements from juveniles to adults. Alternatively, if bone laminarity reflects rapid growth (as has been shown previously in emu), it should be abundant in juveniles and decrease with age as growth slows in adults for all elements. Incorporating growth rate measurements, direct biomechanical data, and direct histological classification of laminarity makes this study a first of its kind that will be able to clarify the importance of growth and mechanics on vascular canal orientation in emu limbs.

Materials & Methods

This study samples forelimb and hindlimb elements from eight emus ranging in age from 2 to 60 weeks (Table 1). Birds used in this study were euthanized as part of a previous study (Main & Biewener, 2007) and the selected elements stored frozen. Emus were originally obtained as hatchlings by R.P. Main (at the time at Harvard University) from commercial farms (Songline Emu Farm, Gill, MA, USA; Scattered Oaks Emu Farm, Iola, TX, USA; Deep Hollow Farm, Oakdale, CT, USA) and raised at Harvard University’s Concord Field Station (Bedford, MA, USA) under Harvard FAS IACUC approval AEP 23–15. For the first eight weeks of life the emus were held in large indoor enclosures, and then moved into pasture-sized outdoor enclosures. All birds had free access to commercial ratite diet (Mazuri, PMI Nutrition International, LLC, Brentwood, MO, USA) and water. Male and female birds were included based on availability. Emus exhibit a minor degree of sexual dimorphism, with females being slightly larger on average (del Hoyo, Elliot & Sargatal, 1992). The difference in size is not large enough to be considered a confounding factor for this study.

Table 1 Emu identification number, age at sacrifice, and mass.

Specimen	Age (weeks)	Mass (kg)	
15	2.3	0.74	
1c	2.4	0.94	
17	4.6	1.53	
14b	8.1	4.73	
16	12	6.85	
2a	15.9	11.13	
21	48	28.9	
23	60.1	29.4	

As a part of a previous study (Main & Biewener, 2007), each bird was given a single intramuscular injection of xylenol orange (80 mg/kg) followed by calcein (30 mg/kg). Injections were given one week apart in birds less than 16 weeks of age and two weeks apart in birds between 16 and 65 weeks of age. Xylenol and calcein are fluorescent labels that are incorporated rapidly into newly mineralizing surfaces of bone at the time of injection (An & Martin, 2003). Thus, the time elapsed and the space between xylenol and calcein labels allows the calculation of periosteal (radial) growth rate. One week after the last injection, surgery was performed to attach strain gauges to the cranial, caudal, and lateral aspects of the left femur and the cranial, caudal, and medial aspects of the left tibiotarsus. Single element strain gauges were used on the lateral femur and cranial and medial tibiotarsus. Rectangular rosette gauges were used on the cranial and caudal femur and caudal tibiotarsus. Rosette strain gauges allow both tensile and compressive principal strains and their orientations to be measured, and were placed so the central element of the gauge was parallel to the long axis of the bone. One day after surgery, the birds were run on a treadmill over a wide range of speeds and gaits. The raw data produced from the strain gauges were converted from voltage to microstrain using a custom MATLAB program. Shear strains were calculated from the rosette strain gauges using standard equations (Biewener & Dial, 1995). High quality shear strain data were most consistently collected from the caudal cortices of the femur and tibiotarsus and that is what is reported here. Trials in which the birds ran with a duty factor near 0.50 are included in the shear strain analysis (mean ± SD: 0.50 ± 0.02). Duty factor is the proportion of the time that the animal’s foot spends on the ground during a stride relative to the entire stride time. A duty factor of 0.50 represents the point at which the birds transition to a running gait that incorporates an aerial phase. This is a relatively slow run for emu, but represents the duty factor for which we could maximize the number of animals included in this study based upon successful strain gauge function. Each trial was represented by five footfalls and, generally, two trials were collected for each bird. Following bone strain data collection, animals were euthanized. After death, whole wings were removed from the individuals and stored frozen. Histological sections of the femora and tibiotarsi were prepared (see (Main & Biewener, 2007 for details) and shipped with the frozen wings to Midwestern University.

Histological preparation

Emu wings were thawed and feathers, skin, muscles, and tendons were reflected to expose the skeletal elements. Both right and left wings were used based on availability. Using digital calipers, total length of each bone was measured and recorded. A 37-mm segment was removed using a Dremel tool from the mid-shaft region of the humerus, ulna, and radius. For two and four week old individuals whole elements were harvested due to their small size. Segments were labeled with permanent marker to maintain orientation. Dissected bone segments were placed in 10% neutral buffered formalin for fixation and then dehydrated in a graded ethanol series (70%, 85%, 100%) under vacuum. Segments were cleared with a xylene-substitute (Histo-clear; National Diagnostics, Atlanta, Georgia, USA). The bone segments were then vacuum- infiltrated and embedded in glass vials using Osteo-Bed Plus Resin, a two-part methyl methacrylate (Polysciences Inc.). Vials were placed in a 32 °C bead bath to fully harden.

Once the resin hardened, vials were broken and two roughly 800-µm transverse sections were cut using a diamond blade saw (Isomet 1000; Buehler, Lake Bluff, Illinois, USA). These sections were attached to frosted glass slides using two-ton epoxy (Devcon, Milpitas, California, USA), keeping consistent spatial orientation. Slides were then ground to a thickness of 100 ± 10 µm using a graded scale of grit paper on a stand grinder (Metaserv 250; Buehler, Lake Bluff, Illinois, USA) and coverslipped with Permount (Fisher Scientific). The histological preparation was modified from An & Martin (2003) and closely followed Lee & Simons (2015).

Image collection

The undecalcified sections contain xylenol (orange) and calcein (green) fluorochromes that were incorporated into newly mineralizing bone at the time of injection (see above for injection schedule). These fluorochromes create stable long lasting tags (Van Gaalen et al., 2010) and were examined under bright-field and fluorescent illumination with a motorized epifluorescent microscope (IX73; Olympus). The xylenol (orange) and calcein (green) tags were revealed using TRITC and FITC filter cubes, respectively, and a multichannel (red, green, bright-field) image of each section was generated with imaging software (cellSens, Olympus). Sufficient optical resolution (10X UPlanAPO ≈0.84 µm; 20X UPlan S-APO ≈0.45 µm) allowed a dual color-monochrome camera (DP80, Olympus) to capture high quality images (10X = 1.02 µm/pixel; 20X = 0.51 µm/pixel).

Figure 1 Representative histological sections of emu femora, tibiotarsi, and humeri from a range of ages.

2.3 weeks (A, B, C), 8.1 weeks (D, E), 12 weeks (F), 16 weeks (G, H, I), 60 weeks (J, K, L) Femora (A, D, G, J), tibiotarsus (B, E, H, K) and humeri (C, F, I, L). Scale bars equal 1,000 µm for femora and tibiotarsi, and 250 µm for humeri. Bright field images of non-straightened caudal or medial octants.

Calculating bone laminarity and radial growth rates

Bright-field and fluorescent images were obtained from the wing and hindlimb elements (Figs. 1 and 2) and divided into equal octants from the estimated bone centroid. Four octants representing the cardinal anatomical positions (wing elements: cranial, caudal, dorsal, ventral; hindlimb elements: cranial, caudal, lateral, medial) were extracted (Fig. 2). Using ImageJ, each extracted octant was then uncurved using the “Straighten” function. The purpose of straightening was to standardize the periosteal tangent line so that appropriate measurements could be made in classifying the orientation of the vascular canals (Lee & Simons, 2015). To ensure there was minimal deformation of the image during the straightening process, known test angles were placed upon the image and measured in relation to the periosteal surface after the straightening function had been applied. Only those images with an average deformation less than or equal to 10° were accepted.

Within each of the four octants, the calcein green and xylenol orange tags were outlined with two reference lines. The distance between reference lines was measured at 10 equally spaced points in each octant. Growth rate was measured by taking the mean distance between consecutive fluorescent tags divided by number of days between injections (Fig. 2C).

Figure 2 Sampling methods for growth rate and laminarity measures.

(A) Gray shading indicates the four octants (wing elements: cranial, caudal, dorsal, ventral; hindlimb elements: cranial, caudal, lateral, medial) sampled on each cross-section. (B) Each octant was isolated, straightened, and a sample area indicated: between fluorochrome reference lines. (C) On fluorescent images, growth rate was measured by taking the mean distance (white arrow) between the periosteal extent of the xylenol (red) and calcein (green/yellow) tags divided by number of days between injections. (D) Laminarity was measured by approximating each in-focus primary vascular canal with the best-fitting ellipse, using the best-fitting ellipse to categorize canal orientations, and calculating the proportion of circumferential canals. (Tibiotarsus 17, 4.6 wks).

Degree of laminarity (Laminarity Index, LI) was measured in the interval of bone between the fluorochrome reference lines across all four octants. Using ImageJ, an ellipse was drawn within each in-focus primary vascular canal in the measurement interval (Fig. 2D). Branching canals were separated at branch points and counted individually. Sharply curving canals were treated as branching. The aspect ratio and angle at which the ellipse sat in relation to the straightened periosteal surface was measured. We used the criteria set forth by de Margerie (2002) to classify the orientation of the vascular canals: (1) “circular” (circumferential) canals are oriented parallel (0° ± 22.5°) to the periosteal surface of the bone; (2) radial canals are orthogonal (90 ± 22.5°) to the periosteal surface; (3) longitudinal canals run parallel to the long axis of the bone and have ellipses with an aspect ratio of less than 3; (4) oblique canals are all other orientations. Only primary vascular canals were measured. Secondary osteons in the sample area were excluded. We used a simple proportion (number of circumferential canals to the total number of canals) to quantify laminarity. To test the growth hypothesis, we used the laminarity index calculated from all sampled octants (Table 2, Table S1). Because consistent high quality shear strain data are only available from the caudal octant, we used the laminarity index calculated from only the caudal octant to test the mechanical hypothesis (Table 3, Table S2).

Table 2 Growth rate and bone laminarity for emu specimens.

Laminarity Index (LI) was measured in a sample area outlined by the periosteal extent of two bone fluorochromes. Forelimb LI was rescaled following (Smithson & Verkuilen, 2006) to prepare the data with 0 values for beta regression.

Specimen	Age (weeks)	Element	Growth rate (µm/day)	LI	Rescaled LI (Forelimb only)	
15	2.3	Femur	130.2	0.03		
Tibiotarsus	62.3	0.02		
Humerus	25.4	0.21	0.22	
Ulna	6.8	0.14	0.16	
Radius	15.2	0	0.02	
1c	2.4	Femur	73.3	0.14		
Tibiotarsus	53.8	0.12		
Humerus	16.2	0.13	0.15	
Ulna	11.6	0	0.02	
Radius	9.5	0.11	0.13	
17	4.6	Femur	162.6	0.02		
Tibiotarsus	99.1	0.01		
Humerus	25.2	0.29	0.30	
Ulna	11.5	0.08	0.10	
Radius	9.4	0.11	0.13	
14b	8.1	Femur	101.1	0.14		
Tibiotarsus	68.5	0.14		
Humerus	23.8	0.33	0.34	
Ulna	14.4	0.19	0.20	
Radius	14.1	0.15	0.17	
16	12	Femur	38.3	0.48		
Tibiotarsus	41.5	0.19		
Humerus	12.2	0.58	0.57	
Ulna	11.8	0.32	0.33	
Radius	8.7	0.27	0.28	
2a	15.9	Femur	29.4	0.56		
Tibiotarsus	29.2	0.35		
Humerus	11.1	0.45	0.46	
Ulna	3.4	0.25	0.26	
Radius	2.4	0	0.02	
21	48	Femur	6.4	0.29		
Tibiotarsus	4.9	0.39		
Humerus	14.6	0.22	0.23	
Ulna	2.2	0.40	0.40	
Radius	1.3	0	0.02	
23	60.1	Femur	5.9	0.51		
Tibiotarsus	3.8	0.58		
Humerus	1.7	0	0.02	
Ulna	1.6	0	0.02	
Radius	1.7	0	0.02	

Robust Principal Components Analysis (RPCA) and beta regression

The explanatory variables thought to affect laminarity in the emu show multicollinearity. For example, mass and age covary with each other (Goonewardene et al., 2003) as well as with growth rate (Montes et al., 2005), and shear strain (Main & Biewener, 2007). If left unaddressed, multicollinearity can decrease precision and reliability when estimating the effect of one variable while holding the others constant (Fekedulegn et al., 2002). Principal components analysis (PCA) accounts for this multicollinearity by forming new uncorrelated variables (i.e., principal components) that are linear combinations of the original explanatory variables while preserving as much of the original variation as possible (Hammer, Harper & Ryan, 2001). However, PCA is highly sensitive to variables with large variances and skewed distributions (Hubert, Rousseeuw & Verdonck, 2009), so we standardized (i.e., centered by the median and scaled by the median absolute deviation) mass, age, growth rate, and shear strain with the function “RobScale” (Signorell, 2019) in R (R Development Team, 2019). This process stabilizes variance and minimizes the effect of absolute scale in the calculation of principal components. Skewed data are often transformed prior to PCA (e.g., logarithmic or Box-Cox), but such transformations may worsen skewness or complicate PCA interpretation (Hubert, Rousseeuw & Verdonck, 2009). Instead, we performed robust PCA (Hubert, Rousseeuw & Verdonck, 2009) as implemented by the R package “rospca” (Reynkens, 2018), which is suitable for skewed data. Three datasets were analyzed separately: (1) cardinal octants from hindlimb elements; (2) caudal octants from hindlimb elements; and (3) cardinal octants from forelimb elements. The results of each robust PCA are presented in Table 4.

Table 3 Caudal shear strain and caudal octant laminarity for emu specimens.

Caudal shear strain data were previously collected by Main & Biewener (2007). Caudal laminarity index (LI) was measured in a sample area outlined by the periosteal extent of two bone fluorochromes in the caudal octant only. LI values were rescaled following (Smithson & Verkuilen, 2006) to prepare data with 0 values for beta regression. Only specimens for which caudal shear strain data were available are included.

Specimen	Age (weeks)	Element	Caudal Shear Strain (microstrain)	Caudal Octant LI	Rescaled Caudal LI	
1c	2.4	Femur	−308	0.21	0.24	
17	4.6	Femur	−1,503	0	0.04	
Tibiotarsus	−1,397	0	0.04	
14b	8.1	Femur	−997	0.05	0.08	
Tibiotarsus	−261	0.06	0.10	
16	12	Femur	−1,491	0.26	0.28	
Tibiotarsus	−947	0.22	0.24	
2a	15.9	Femur	−1,620	0.45	0.46	
Tibiotarsus	−293	0.59	582	
21	48	Femur	−1,657	0.15	0.18	
Tibiotarsus	−1,318	0.55	0.54	
23	60.1	Femur	−2,283	0.53	0.53	

Table 4 Summary of principal components analyses.

Robust principal components analysis summary for (A) cardinal octants from femur and tibiotarsus, (B) caudal octants from femur and tibiotarsus, and (C) cardinal octants from humerus, ulna, and radius.

	PC 1	PC 2	PC 3	
A. Cardinal octants from femur and tibiotarsus	
Eigenvalues	8.182	0.403	0.037	
Standard deviation	2.860	0.635	0.192	
Proportion of variation	0.949	0.047	0.004	
Cumulative Proportion	0.949	0.996	1.000	
Mass	0.560	0.021	0.828	
Age	0.769	0.357	−0.529	
Growth rate	−0.307	0.934	0.184	
	PC 1	PC 2	PC 3	PC 4	
B. Caudal octants from femur and tibiotarsus	
Eigenvalues	9.435	0.989	0.233	0.032	
Standard deviation	3.072	0.994	0.482	0.179	
Proportion of variation	0.883	0.093	0.022	0.003	
Cumulative Proportion	0.883	0.975	0.997	1.000	
Mass	0.520	−0.178	0.098	0.830	
Age	0.804	0.019	0.266	−0.532	
Shear strain	0.185	0.898	−0.381	0.122	
Growth rate	−0.221	0.403	0.880	0.121	
	PC 1	PC 2	PC 3	
C. Cardinal octants from humerus, ulna, and radius	
Eigenvalues	7.977	0.187	0.039	
Standard deviation	2.824	0.433	0.197	
Proportion of variation	0.972	0.023	0.005	
Cumulative Proportion	0.972	0.995	1.000	
Mass	0.563	0.074	0.823	
Age	0.786	0.259	−0.561	
Growth rate	−0.255	0.963	0.087	

For each dataset, the minimum number of principal components (PCs) was selected to cover approximately 95% of the observed variance of the original explanatory variables. We assessed the relationship between PC(s) and mean laminarity index (LI) using beta regression as implemented by the R package “gamlss” (Rigby & Stasinopoulos, 2005). This method is appropriate when the response variable (LI) is a proportion (Warton & Hui, 2011). To accommodate values of 0 in the caudal hindlimb and cardinal forelimb datasets, LI values were rescaled to the effective interval of [0.005, 0.995] (Smithson & Verkuilen, 2006). The logit link function was used to connect mean LI to a linear combination of the PCs. Pseudoreplication is a concern because different bone elements were sampled from the same individual (Hurlbert, 1984; Gillies et al., 2006; Lee & O’Connor, 2013; Jordan, 2018), so we combined the logit link function with a random-intercept model as follows: (1) μlogitLI=β0+β1PC1+β2PC2+β3Element+γ

where PC is the principal component, β is regression coefficient, Element is a dummy variable coding for element type, γ is the random-intercept effect of “specimen ID”, and μlogit(LI) is the logit link function for the mean of LI (Ferrari & Cribari-Neto, 2004).

For each cardinal dataset, we evaluated two models. The first model includes PC 1 as the sole predictor given that it accounted for approximately 95% of the variance in the original explanatory variables. The second model adds element type as a dummy variable. For the caudal dataset, PC 1 and PC 2 covered at least 95% of the variance in the original explanatory variables. Therefore, we evaluated six models. The first three models involve PC 1 and PC 2 individually as sole predictors as well as together in additive combination. The remaining three models add element type as a dummy variable (Table 5). The small-sample correction of Akaike’s Information Criterion (AICc) (Hurvich & Tsai, 1989) was used to compare models within each dataset. In general, the best supported model has the lowest AICc value (Burnham & Anderson, 2002). Relative support between the best and alternative models was assessed with difference (ΔAICc) values. Alternative models with ΔAICc values greater than 3, which is equivalent to a p-value of 0.051 (Taper, 2004), were rejected as having weak support. Raw data and R script for analyses can be found in the Supplementary Files (Tables S1, S2, Code S1).

Table 5 Comparison of random-intercept beta regression models.

Model selection was based on Akaike’s Information Criterion (AICc) value and ΔAICc. (A) Cardinal octants from hindlimb, (B) caudal octants from hindlimb, and (C) cardinal octants from forelimb. For each dataset, the best supported model showed the “ontogenetic axis (PC 1) as the sole predictor of laminarity.

	Model 1	Model 2	
Variable	β	p-value	β	p-value	
A. Cardinal octants from hindlimb	
Intercept	−1.120	1.25e−5	−1.053	3.19e−4	
Element (TBT = 1)			−0.309	0.154	
PC 1	0.337	5.51e−5	0.348	8.92e−5	
Pseudo R2		0.889		0.913	
AICc		0.5		20.7	
ΔAICc		0		20.2	
	Model 1	Model 2	Model 3	
Variable	β	p-value	β	p-value	β	p-value	
B. Caudal octants from hindlimb	
Intercept	−0.970	0.001	−1.105	0.002	−1.083	0.002	
PC 1	0.186	0.013			0.242	0.007	
PC 2			−0.200	0.169	−0.356	0.056	
Pseudo R2		0.775		0.841		0.828	
AICc		19.1		51.5		41.4	
ΔAICc		0		32.4		22.3	
	Model 4	Model 5	Model 6	
Intercept	−1.214	0.006	−1.243	0.011	−1.110	0.011	
Element (TBT = 1)	0.508	0.118	0.432	0.224	0.092	0.782	
PC 1	0.215	0.013			0.242	0.017	
PC 2			−0.015	0.911	−0.319	0.135	
Pseudo R2		0.875		0.882		0.837	
AICc		91.0		181.6		107.9	
ΔAICc		71.9		162.5		88.8	
	Model 1	Model 2	
Variable	β	p-value	β	p-value	
C. Cardinal octants from forelimb	
Intercept	−1.508	1.42e−7	−0.997	7.57e−5	
Element (Radius = 1)			−1.345	6.77e−4	
Element (Ulna = 1)			−0.601	0.044	
PC 1	−0.112	0.089	−0.117	0.030	
Pseudo R2		0.277		0.704	
AICc		−26.6		−20.6	
ΔAICc		0		6	

Results

Measured growth rates ranged from 1.3 µm/day (radius of 48-week-old individual) to 162.6 µm/day (femur of 4.6-week-old individual) (Table 2). Laminarity indices from cardinal octants ranged from 0 to 0.58 (Table 2, Table S1, Fig. 3). Laminarity indices from the caudal octant ranged from 0 to 0.81 (Table 3, Table S2).

Figure 3 Laminarity Indices for the forelimb and hindlimb bones included in this study.

Laminarity indices (LI) measured from the cardinal octants for each bone (Fem = femur, Tbt = tibiotarsus, Hum = humerus, Ulna, Rad = radius) from specimens (A) 15–2 week, (B) 1c–2 week, (C) 17–5 week, (D) 14b–8 week, (E) 16–12 week, (F) 2a–16 week, (G) 21–48 week, and (H) 23–60 week. LI was measured between fluorochrome reference lines on all sampled octants.

Regression analysis of cardinal octants from the femur and tibiotarsus

Principal component (PC) 1 (eigenvalue = 8.181), consisting of mass, age, and growth rate, accounts for 95% of the cumulative variance (Table 4). Mass and age loadings have the same sign, whereas growth rate loading has an opposite sign. The loadings suggest that PC 1 represents an “ontogenetic axis” with juvenile features (small size, young age, and rapid growth rate) at one end and adult features (large size, old age, and modest growth rate) at the other (Fig. 4). PC 2 (eigenvalue = 0.403) consists of the residual variation in (i.e., “ontogeny-independent”) growth rate and accounts for 4.7% of the cumulative variance (Table 4).

Figure 4 Robust principal components analysis of the cardinal octants from the femur and tibiotarsus.

An “ontogenetic axis” (PC 1) accounts for 95% of the variance with juvenile features to the left (small size, young age, and rapid growth) and adult features to the right (large size, old age, and modest growth). Residual variation in growth rate is absorbed into PC 2, which accounts for another 4.7% of the cumulative variance.

The random-intercept beta regression model without element type as a predictor has overwhelming support (Table 5). It predicts that ∼89% of the variation in laminarity is explained by the “ontogenetic axis” (Table 5). Juvenile features (e.g., small size, young age, and rapid growth rate) are correlated with lower laminarity values, whereas adult features are correlated with higher laminarity values (p < 5.51e−5; Fig. 5).

Figure 5 Effect of the “ontogenetic axis” (PC 1) on laminarity (LI) in the cardinal octants of the hindlimb.

Random-intercept beta regression reveals that the “ontogenetic axis” (PC 1) accounts for 89% of the variation in laminarity from the cardinal octants of femur (green triangles) and tibiotarsus (blue squares). Elevated laminarity values are strongly correlated with adult features such as large size, old age, and modest growth rate (p < 5.51e−5).

Regression analysis of caudal octants from the femur and tibiotarsus

Table 4 shows that the first two PCs account for at least 95% of the cumulative variance—88% by PC 1 (eigenvalue = 9.435) and 9% by PC 2 (eigenvalue = 0.989). Similar to cardinal octant data, mass, age, and growth rate contribute strongly to PC 1. Their loadings are also consistent with variation along an “ontogenetic axis” with juvenile features at one end (small size, young age, and rapid growth rate) and adult features at the other (large size, old age, and modest growth rate). Although strain has a minor contribution to PC 1, it dominates PC 2, which we interpret as a “loading effect axis” (Fig. 6).

Figure 6 Robust principal components analysis of the caudal octants from the femur and tibiotarsus.

An “ontogenetic axis” (PC 1) accounts for 88% of the variance with juvenile features to the left (small size, young age, and rapid growth) and adult features to the right (large size, old age, and modest growth). PC 2 explains 9% of the cumulative variance and is dominated by shear strain, forming a “loading-effect axis.”.

The model with PC 1 as the sole predictor of caudal octant laminarity has strongest support based on ΔAICc and explains 78% of the variation in caudal octant laminarity (Table 5). Caudal and cardinal datasets show slightly different estimates for the coefficient of PC 1, which is consistent with slight inter-octant variation in laminarity. Variation aside, the overall ontogenetic trend is similar: higher laminarity values are correlated with adult features, whereas lower laminarity values are correlated with juvenile features (Fig. 7; p = 0.013). Although shear strain contributes to PC 1 and generally increases along the “ontogenetic axis”, the effect is relatively weak. To highlight this, we multiplied the regression coefficient and eigenvectors of PC 1. The resulting standardized coefficients of the original predictors (growth rate = − 0.041; mass = 0.097; age = 0.150; strain = 0.035) suggest that the relative effects of growth rate, mass, and age on caudal octant laminarity in the hindlimb are 1.2 to 4.3 times greater than that of strain.

Figure 7 Effect of the “ontogenetic axis” on laminarity (LI) in the caudal octants of the hindlimb.

Random-intercept beta regression reveals that the “ontogenetic axis” (PC 1) accounts for 78% of the variation in laminarity from the caudal octants of femur (green triangles) and tibiotarsus (blue squares). Laminarity generally increases along that axis (p = 0.013) to which shear strain has a minor contribution.

The caudal hindlimb dataset does not support the “loading effect axis” (PC 2) as a strong predictor for laminarity. Compared to model 1 (PC 1 as the sole predictor), the remaining alternative models with PC 2 only explain an additional 5–6% of the variation in caudal hindlimb laminarity. Each of these alternative models has ΔAICC larger than 3 indicating weak to no support (p > 0.05). Furthermore, the model coefficient for PC 2 in model 2 (sole predictor) and model 3 (additive combination with PC 1) is not significant (p-value equals 0.169 and 0.056, respectively; Table 5).

Regression analysis of cardinal octants from the humerus, ulna, and radius

In the absence of strain data for the forelimb elements, mass, age, and growth rate dominate PC 1 (eigenvalue = 7.977), and this “ontogenetic axis” accounts for 97% of the cumulative variance (Table 4, Fig. 8). PC 2 (eigenvalue = 0.187) absorbs residual variation in growth rate and accounts for an additional 2.3% of the cumulative variance.

Figure 8 Robust principal components analysis of the cardinal octants from the humerus, ulna, and radius.

An “ontogenetic axis” (PC 1) accounts for 97% of the variance with juvenile features to the left (small size, young age, and rapid growth rate) and adult features to the right (large size, old age, and modest growth rate). Residual variation from growth rate largely contributes to an “ontogeny-independent growth rate axis” along PC 2. This axis only explains 2.3% of the cumulative variance.

Although the model with “ontogenetic axis” (PC 1) as the sole predictor has the best support, it only accounts for 27.7% of the variation in forelimb laminarity. Moreover, the model coefficient for PC 1 is not significant (p = 0.089; Table 5).

Discussion

Does bone laminarity reflect fast growth?

The highest periosteal growth rate in all elements was found in the femur of the 4.6-week old individual (Table 2). As expected, hindlimb elements had higher growth rates than forelimb elements, reaching a maximum of 163 µm/day in the femur and 99 µm/day in the tibiotarsus. The humerus grew the fastest of the wing elements, reaching a maximum rate of 25 µm/day measured in the 2.3 and 4.6 week old individuals. Birds older than 8 weeks experienced a drastic decrease in bone growth rate in both hindlimb and forelimb elements. Previous analysis of emu somatic growth rate (increase in body mass) showed the maximum rate of growth (inflection point) to be about 15–17 weeks of age (Goonewardene et al., 2003). Our results reveal that age at maximum bone growth (approximately 5 weeks) precedes the somatic growth inflection, similar to other vertebrates (Lee & O’Connor, 2013). Therefore, caution is warranted when inferring somatic life-history milestones, such as growth spurts, solely from skeletal data.

Principal components analysis reveals a large proportion of variance lies along an “ontogenetic axis” (Figs. 4, 6 and 8). One end of the axis is represented by juvenile traits such as small size, young age, and rapid growth, whereas adult traits such as large size, old age, and modest growth characterize the other end. This “ontogenetic axis” has a significant influence on laminarity in the femur and tibiotarsus (Figs. 5 and 7), whether analyzed in cardinal octants (p < 0.001) a single octant (p = 0.013). Elevated laminarity in the hindlimb appears correlated with adult features, including modest growth rate. This relationship is consistent with findings in the king penguin that also reported laminar bone to be associated with modest growth rates in four limb bones: femur, tibiotarsus, humerus, and radius (de Margerie et al., 2004). More recently, a study using microCT to assess three-dimensional vascular canal orientation in the humerus and femur of growth-controlled broiler chickens also found elevated laminarity in a slow-growing (feed-restricted) group (Pratt & Cooper, 2018). Interestingly, the effect of the “ontogenetic axis” on laminarity in the wing elements is weak (p = 0.089), only explaining 27.7% of variation (Fig. 9). The weakened “ontogenetic axis” in wing bone laminarity is consistent with relaxed selection on the vestigial wing leading to increased anatomical variability in the species (Maxwell & Larsson, 2007).

Figure 9 Effect of the “ontogenetic axis” on laminarity (LI) in cardinal octants of the forelimb.

Random-intercept beta regression reveals poor correlation (p = 0.089) between laminarity and the “ontogenetic axis” in the humerus (purple triangles), ulna (red squares), and radius (orange circles).

Notably, our results differ from those previously reported for young emu bones in which laminar and reticular bone was found in the fastest growing hindlimb bones (Castanet et al., 2000). In particular, Castanet et al. (2000) found laminar bone to be most abundant in the femur and tibiotarsus of emu less than 2 months of age, which corresponds to the youngest individuals in our study. Based on reported body masses, the emus included in our study were about 2–3 times heavier than the emus in the Castanet et al. (2000) study for a given age (Table 1). The reason for the differences in size and ontogenetic patterns for bone vascularity types between these two emu samples remain unknown, but could be related to genetic, dietary, or rearing conditions between the two groups. If laminarity is associated with lower growth rates, the youngest emus we studied may have been growing too fast for laminar bone to form. The highest growth rate measured was in the femur of the 4.6-week-old bird (162.62 µm/day), which was about twice the highest growth rate found in the femoral reticular bone tissue reported in the prior study (89.4 µm/day). Our study did not specifically address reticular bone, but by taking the proportion of oblique vascular canals (a “reticular index”), we found the amount of reticular bone in the fastest growing individual to be low in the hindlimb elements (femur and tibiotarsus: 0.17), and moderate to high in the wing elements (humerus: 0.62, ulna: 0.58, radius: 0.45). This result is, at least, consistent with the previous study because Castanet et al. (2000) found reticular bone to be more abundant in the humerus. The data analysis in the earlier emu study (Castanet et al., 2000) was conducted before a more rigorous method for quantifying canal orientation was developed (de Margerie, 2002), which may contribute to the observed differences between the two separate emu populations. Regardless, the results of the present study do not support the hypothesis that elevated laminarity reflects rapid growth in the emu and instead link elevated laminarity with more modest growth rates.

Does bone laminarity reflect biomechanical load?

Robust PCA is consistent with previous scaling analysis showing that shear strains in the femur and tibiotarsus of the emu increase with growth (Main & Biewener, 2007). Shear strain covaries with mass, age, and growth rate along an “ontogenetic axis” such that juveniles experience smaller shear strains in the caudal octants, whereas adults experience larger strains (Table 4; Fig. 6). However, shear strain contributes comparatively less to the “ontogenetic axis” than the other covariates. Consequently, it has a correspondingly minor effect on caudal octant laminarity in the hindlimb. Indeed, transformation of the effect of the “ontogenetic axis” on laminarity back into the scale of the actual covariates reveals that the standardized effects of age (0.150), mass (0.097), and growth rate (−0.041) are relatively larger than that of shear strain (0.035). Elevated laminar bone tissue, in combination with increased bone mineralization and decreased bone curvature during growth, may have collectively helped mitigate shear strains despite the large increase in mass (Main & Biewener, 2007).

Residual variation in shear strain that is not accounted for by the “ontogenetic axis” forms a “loading effect axis” (Table 4; Fig. 6). However, this effect does not correlate with laminarity either as the sole predicator (p = 0.169) or in additive combination with ontogeny (p = 0.056). Put together, the results clearly show that ontogenetic factors largely influence the formation of laminar bone in the caudal hindlimb of the emu, although torsion-induced shear strain is a minor additional factor.

The weak association between laminarity and shear strain limits the predictive potential of this relationship. Our results for emu hindlimb bones are consistent with previous studies of other limb elements presumably loaded in torsion. For example, when comparing wings of similar shape, laminarity in wing bones can be similar despite differences in presumed biomechanical load associated with unique primary flight modes (Simons & O’Connor, 2012; Marelli & Simons, 2014). In addition, preferred flight mode may only have subtle effects on overall loading of the bones, with the dominant loads being the high strains present during take-off (Biewener & Dial, 1995). Furthermore, despite sharing with birds convergent features related to powered-flight such as torsionally loaded bone with relatively thin cortical walls (Swartz, Bennett & Carrier, 1992), bone mineral density (Dumont, 2010), and metabolic rate (e.g., Maina, 2000), bats lack laminar bone entirely (Lee & Simons, 2015; Pratt et al., 2018). Instead, they have bones that are poorly-vascularized with a parallel-fibered matrix. Because that histology tends to grow very slowly in other species (e.g., <five µm d−1 de Margerie, Cubo & Castanet, 2002; Castanet et al., 2004), Lee & Simons (2015) speculated that bats do not grow fast enough to form laminar bone. Indeed, somatic growth is approximately four times slower in bats than in birds of comparable size (Lee & Simons, 2015). Caution is warranted, however, as we show in the emu that interchanging somatic and skeletal growth may be misleading. Whatever the actual cause is, the evidence is clear that elevated laminarity is not a prescriptive feature of torsionally loaded bone.

In this study, age groups are represented by one individual, with the exceptions of the youngest and oldest age groups that contain two. A larger sample of individuals in each age group would allow for investigation of how individual variation may or may not affect the relationship between LI, shear strain, and growth rate. Laminarity indices can be quite variable among individuals in some species. The pigeon humerus, which has been shown to experience large torsional loads, has been documented to exhibit both high and low laminarity in different individuals (Lee & Simons, 2015; Ourfalian, Ezell & Lee, 2016; Skedros & Doutré, 2019). Similarly, a pooled sample of humeri from eight Red-tailed hawks show LI values that range from 0.30–0.70 (Simons & O’Connor, 2012; Marelli & Simons, 2014). Whether these variability patterns are biological or methodological is unclear. Laminarity measured on a histological section is a 2-dimensional representation of a 3-dimensional meshwork of vascular canals in cortical bone. This research is limited by the assumption that one or two closely placed mid-shaft histological sections are an accurate representation of vascular canal structure. MicroCT-based assessment of the three-dimensional network of vascular canals suggests that traditional 2D histological methods may overestimate LI, but also recognizes that these differences may be methodological (Pratt & Cooper, 2017; Pratt et al., 2018). Certainly, future studies should continue to use microCT to assess how well laminarity measured on histological sections represents actual biological structure. In addition, the torsional resistance in bones may more likely be linked to the specific orientation of another histological feature: collagen fibers. Collagen fiber orientation (CFO) has been shown to reflect principal strain distributions (Riggs, Lanyon & Boyde, 1993; Skedros & Hunt, 2004; Skedros, Hunt & Bloebaum, 2004; Skedros & Doutré, 2019). Analysis of CFO is beyond the scope of the current study. However, given the known positive correlation between transversely oriented collagen fibers and bone laminarity (de Margerie et al., 2005), we would expect a similar pattern for the femora and tibiotarsi examined here.

Although there is no direct biomechanical data for the forelimb elements of these birds, the wing elements presumably experience minimal loading. The emu wing is extremely reduced in size, even when compared to other ratites, and has almost no observed function (del Hoyo, Elliot & Sargatal, 1992). Wing muscles of emu contain primarily slow acting tonic muscle fibers that may not allow much wing movement (Maxwell & Larsson, 2007), which suggests the underlying wing elements would experience minimal biomechanical loading. Despite the assumption that the emu wing is under minimal load, a moderate to high degree of laminarity was found in at least the humerus and ulna (Table 2). This laminarity can be attributed to the modest bone growth rate observed in the wing elements and/or to the third factor affecting bone microstructure: phylogenetic relationships. Within the paleognaths, it has been hypothesized that at least three independent flight losses have occurred, with only one order (the tinamous) still retaining the ability to fly (Harshman, Braun & Braun, 2008; Mitchell et al., 2014). The moderate/high wing bone laminarity may be a feature of the flighted common ancestor of paleognaths that is retained in the flightless descendants. Indeed, significant phylogenetic signal has been found in some osteohistological features in a sample of paleognaths (Legendre et al., 2014). Future studies should investigate the histological and in vivo loading of the flighted relatives of emus to better understand the potential influence of phylogeny on bone laminarity.

Conclusions

In the emu limb skeleton, ontogenetic factors such as size, age, and growth rate have major effects on vascular canal orientation. The effect of shear strain is relatively weak and suggests that laminar bone is not a good predictor of torsional loading. Even though the forelimb elements likely experience minimal loading, the humerus and ulna show wide variation in laminarity, perhaps due to relaxed selection. Future studies should investigate laminarity in other palaeognathous birds to better understand the effects phylogeny, ontogeny, and torsional loading have on bone laminarity. Other future work should focus on the experimental manipulation of biomechanical loads to observe the effects on vascular canal orientation in limb bones and to better understand to what extent torsional load influences the development of limb bone laminarity. It is also important that variation found between different populations be addressed and studied further. Emu body mass growth rates vary among populations (e.g., Goonewardene et al., 2003), but it is unknown to what extent laminar bone also varies with environmental, dietary, or genetic factors. This study has shown that in emu limb bones, laminarity reflects a complex interplay of ontogeny and biomechanical loads.

Supplemental Information

Table S1 Number of vascular canals in each orientation category (circular, radial, oblique, and longitudinal) counted between fluorochrome reference lines

Vascular canals were counted in a sample area outlined by the periosteal extent of two bone fluorochromes in four octants representing the cardinal anatomical positions (wing elements: cranial, caudal, dorsal, ventral; hindlimb elements: cranial, caudal, lateral, medial). Laminarity Index (LI) measures the proportion of circular to total number of canals.

Click here for additional data file.

Table S2 Number of vascular canals in each orientation category (circular, radial, oblique, and longitudinal) counted between fluorochrome reference lines

Vascular canals were counted in a sample area outlined by the periosteal extent of two bone fluorochromes in the caudal octant only. Laminarity Index (LI) measures the proportion of circular to total number of canals.

Click here for additional data file.

Code S1 R script to replicate analyses

Click here for additional data file.

Supplemental Information 1 Specimen numbers from the Paleohistology Repository

Click here for additional data file.

The authors would like to thank K Ezell for assistance with histological preparation and imaging. We are grateful for free access to the microscopes in the Microscopy Core Facility at Midwestern University. We thank Andrew Biewener and the graduate students and postdocs at the Concord Field Station (2002-2006) for assistance with bone strain data collection. We thank reviewers Edina Prondvai and Jorge Cubo for useful comments that improved the manuscript.

Additional Information and Declarations

Competing Interests

Author Contributions

Animal Ethics

Data Availability

Data Deposition

The authors declare there are no competing interests.

Amanda L. Kuehn performed the experiments, prepared figures and/or tables, authored or reviewed drafts of the paper, approved the final draft.

Andrew H. Lee conceived and designed the experiments, analyzed the data, contributed reagents/materials/analysis tools, prepared figures and/or tables, authored or reviewed drafts of the paper, approved the final draft.

Russell P. Main conceived and designed the experiments, performed the experiments, contributed reagents/materials/analysis tools, authored or reviewed drafts of the paper, approved the final draft.

Erin L.R. Simons conceived and designed the experiments, performed the experiments, contributed reagents/materials/analysis tools, prepared figures and/or tables, authored or reviewed drafts of the paper, approved the final draft.

The following information was supplied relating to ethical approvals (i.e., approving body and any reference numbers):

Emu strain gauge studies were approved by Harvard FAS IACUC (AEP 23-15).

The following information was supplied regarding data availability:

The raw vascular canal counts used to calculate laminarity are available in the Supplemental Tables. The script is available in Code S1. The specimens are being stored frozen at Purdue University under specimen numbers 15, 1c, 17, 14b, 16, 2a, 21, 23.

Specimens have been deposited at the Paleohistory Repository; accession numbers can be found in the Supplemental File.

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
