# Peer review of "The effects of growth rate and biomechanical loading on bone laminarity within the emu skeleton"

_PeerJ, doi:10.7717/peerj.7616_

## Round 0.1 · original submission · Major Revisions

The two reviewers still find plenty of value in this study and it certainly has improved.The reviewers have been diplomatic and transparent, in my view. They (especially R1) have numerous remaining concerns, which generally add up to moderate revisions that should be required unless there are clear reasons why certain revisions should not be done, in which case the Rebuttal document should explain that along with all other points. Thank you again for this nice study, which will be re-reviewed but I can see that it is progressing toward a positive outcome.

·

Basic reporting

Even though crucial for this study, discussion of Lee & Simons, 2015 and Pratt et al. 2018 findings which both failed to find support for torsional resistance role in bone laminarity in birds (and bats) is largely (or entirely) missing from the MS. In case of Pratt et al. 2018, this can probably be attributed to this paper being comparatively new. However, it is surprising that the findings of Lee & Simons 2015 on bird bones, with both authors being coauthors on the current manuscript, is not regarded in depth. In any case, both papers are of crucial importance for the current study, and hence their findings largely contradicting the conclusions of the current study must also be addressed and included in the discussion.

Experimental design

A figure or an interpretative drawing of the cortex sampling areas, i.e. octants and placement of the sampling point and sample boxes shown on two representative bones for growth rate and biomechanics tests, respectively, would be needed for the readers to get a better visual idea, even if this figure is added as supplementary only.

Statistical procedures are not well documented and there is likely a terminological confusion between logistic regression vs linear regression on log-transformed data, but due to insufficient provision of methodological details of their statistical analyses I cannot be sure.
However, if the authors really meant “logistic regression” with application of the Wilson estimate for confidence intervals, then I cannot see how their data structure was suitable for performing such an analysis.

As far as I could look it up in statistics, logistic regression is a probability regression with binary response variables and its graphical representation is the classical S-shaped curve with an inflection point at 50% probability. The binary nature of observations in logistic regression is expressed as "failure" or "success", and the probability comes from an x times success out of n times try. The same goes for the Wilson estimate: it's a method for binomial proportional confidence interval calculation. So how could the authors apply this method on their laminarity index (LI) data which, even if ratio values, still represent continuous variables and not binary? I could see this work only if they e.g. define "success" as having LI above a certain level, and failure as finding LI below that level; this would provide the binary data structure required for logistic regression (i.e. proportion of LI above vs below the specified level).

However, the provided dataset and the resulting figures do not support such a data analysis setup, so I’m wondering whether the authors actually meant that they log-transformed the dependent (LI) or both, dependent and independent variables and did a simple linear regression on the transformed data? If yes, this must be corrected throughout the MS. Otherwise, providing details of how (and why) they could use a logistic regression for their non-binary data is absolutely necessary.

In any case, an additional table summarizing test statistics and their results would also be necessary.

In addition, the function for fitting the trend lines in Figs 3 - 5 is also missing and should be specified.

Validity of the findings

My concerns about their statistical approach is mentioned in the "experimental design" evaluation field. All uncertainties originating from the concerned analysis of the data may affect the results and conclusions based on them, so without the statistical procedure properly laid down I cannot judge whether their conclusions are supported by their data. Nevertheless, based on their plots of raw data (Fig 3-5), their major conclusions on the relationship between growth rate and laminarity, and between shear strain and laminarity seem to hold true.

Additional comments

I find this work very interesting and would like to encourage the authors for doing more research in this exciting direction, and once sufficient and robust dataset was collected and analysed on extants, to extrapolate their findings on fossil dinosaur - bird transitional taxa (as the authors themselves also pointed out in their MS).

However, besides the comments mentioned in the major 3 fields, I'd like to point out some smaller issues that should also be addressed for scientific completeness.

In Results:
1) Figure 4 shows fewer data points used to construct the fitted curve than actually available for wing bones based in Table 2.

2) A simple plot of laminarity indices including all studied elements per specimen for visual comparison of differences among elements and age categories would be very useful and informative about general vascular pattern distribution within the skeleton and during ontogeny.

3) Surprisingly, spatial distribution of different vascular orientations throughout the cortex is not reported here to any extent. A potentially consistent regional difference in vascular architecture with homologous cortical areas showing similar orientations but a high variance among different areas could result in high overall cortical variance in LI. However, this should be distinguished from high variance in LI originating from more irregular orientations with no location-specific patterning throughout the entire cortex. This would also be very important to look at from an evolutionary perspective, as one would expect that low to no functional selection pressure on wing bones would result in high and random variance in its histological features which would have to be clearly distinguished from a potentially high overall variance with regionally consistent patterning.

In Discussion
1) Some aspects and questions are left open. For instance, even if non-significant, there is a clear negative trend between growth rate and laminarity in all wing bones too, based on Figure 5, but this is not considered in the discussion.

2) They also mention a three times difference in recorded mass growth rates of emus between their study and Castanet et al’s (2000) but there is no detail on how they assessed growth rates in body mass for their own specimens and whether their method to do so is the same as in Castanet et al (2000). Furthermore, they don’t consider possible reasons for this significant difference in mass growth rates. E.g. was there any difference in the diet, climatic conditions, locomotor freedom, etc under which emu specimens used in the two different studies were kept? Or were they a different breed? This is important, as these unknown factors may not only influence growth rates, but could partially be directly responsible for the differences detected in the prevalence of bone laminarity.


You find further detailed comments in the attached annotated pdf.

In sum, I would recommend publication of this work after moderate/major revision.

Sincerely,
Edina Prondvai

·

Basic reporting

No comment

Experimental design

No comment

Validity of the findings

No comment

Additional comments

Dear authors
The Ms entitled "The effects of growth rate and biomechanical loading on bone laminarity within the emu skeleton" is an important contribution to our knowledge of the functional significance of bone histological variation.
In my view, authors have to deal with 3 issues in order to improve the quality of the Ms.
1. Biomechanics. Skedros and Hunt (2004) tested in the turkey ulna diaphysis whether the degree of laminarity correlate with site-specific differences in collagen fibre orientation in primary bone. They showed highly significant positive correlations between laminarity index and predominant collagen fiber orientation. This relevant contribution is not cited in the Ms under revision ! According to an increasing body of knowledge, it seems that the histological feature linked to (explaining) bone biomechanical properties is the orientation of collagen fibers, the orientation of vascular canals being an indirect proxy of them. Moreover, collagen fiber orientation can be conserved in the fossil record during hundreds of million years, allowing paleobiological inferences of biomechanical features. Why did you analyze vascular orientation instead of collagen fiber orientation?
2. Growth rate. According to the results obtained by the research team of Marotti (see for instance the review of Marotti, 2010), static ossification produces woven bone at high growth rate whereas dynamic ossification produces parallel fibered bone or lamellar bone at low growth rates. It seems that collagen fiber orientation (rather than vascular canals orientation) is linked to bone growth rate. Again, why did you analyze vascular orientation instead of collagen fiber orientation?
3. Integrative analyses. Authors analyzed separately the relationships between, on the one hand, bone growth rate and bone laminarity and, on the other hand, shear strain and bone laminarity. However, the explanatory variables bone growth rate and shear strain interact among them and the effect of this interaction on the response variable (bone laminarity) has to be taken into account using for instance multiple regression. Moreover, the effects of body mass and age must also be taken into account in these integrative analyses.
Please don’t hesitate to contact me for any clarification.
Yours sincerely
Jorge Cubo

---

## Round 0.2 · Minor Revisions

One reviewer has opted out from further review but the 2nd reviewer has provided further comments, which are quite constructive and careful, so I will refrain from bringing in a new reviewer. I agree that the full R code and data should be provided with the study (in Supp Info or on repository such as Figshare etc.) so that the data can be checked/used by others in the future; that is in keeping with the open science nature of PeerJ. Your very attentive revisions in last round were much appreciated, and we look forward to the newly revised MS. Further review will not be needed if the guidelines here are adhered to and the reviewer's comments are fully addressed to our satisfaction.

·

Basic reporting

As far as I’m concerned, the revised manuscript of Kuehn et al. satisfies every point of the criteria of basic reporting set by Peer J.

Experimental design

I am not a statistician but as much as I can judge, the new statistical approach applied in the revised version is appropriately chosen for testing their hypotheses and analyzing such data.

However, I still have a question concerning the implied necessity of log 10 transformation of original explanatory variables before turning them into PCs, and some missing details that should be added for a comprehensible methodology.

The authors say: “To minimize this multicollinearity, the variables of mass, age, growth rate and shear strain were normalized with log10-transformation and transformed to principal components”

As I understand, it is frequently stated that PCA can be used to combine highly correlated predictor variables to resolve multicollinearity, and that's why you used it. Besides that multicollinearity is not really minimized but rather preserved / combined in the PCs, the current formulation sounds like normalization via log 10 transformation would also be a necessary step on the way to minimize multicollinearity among your different predictor variables, which is, I assume, not the case.
Also, whether log 10 transformation actually normalizes the data or not depends on the original data distribution because only certain distribution types, mostly log-normal distribution, can be normalized by log transformation. (see e.g. Feng et al. 2014). Did data distribution for all your variables indeed become normal after log 10 transformation?

Furthermore, although I didn’t find it specifically stated in the MS whether you used correlation or covariance matrix for your PCA (which should be stated), if you used correlation matrix which, concerning your differently ranged and scaled variables, is needed here, it already includes a transformation, i.e. standardization to mean = 0 and standard deviation = 1. So why would you use an extra, log 10 transformation as input for PCA?

In sum, I don't understand why this log 10 transformation is necessary here? As much as I know, PCA does not require normally distributed data, and collinearity, which has a significant impact on the outcome of the PCA as well, cannot be canceled by log transformation of input variables. I may be wrong with all this, but I don’t think that log 10 transformation will ensure normal distribution of the resulting PC scores, either, and that the following beta regression, being an ML regression, requires normally distributed explanatory variables.
Besides this, such data transformations usually make biological interpretation of the analysis results as well as the relationships in the original dataset more difficult.

Please, explain why this step was deemed necessary and included in the analysis. On the other hand, if it wasn’t necessary, I suggest you use raw data of predictor variables as an input for a correlation matrix-based PCA. This would allow a better interpretation of the end results too, even if it does not change the outcome drastically (which it probably won’t but it may change significance levels of regressions).

Also, please provide R script and maybe also R output in SI showing parametrization and link functions used in your beta regression models, or alternatively add an SI table summarizing this information. As this method is not frequently applied in such studies, this information would help readers tremendously to comprehend and later also apply your procedure.

Validity of the findings

The results gained with the new statistical method and their interpretations are sound, their logical connection with the hypotheses raised are clear, and now are nicely put in the necessary context of previous studies.
Although I suggested that the analyses are rerun with raw data instead of log 10 transformed predictor variable values, if implemented, the main conclusions of this investigation will most likely not change (as opposed to the previous MS version which gave quite different results and interpretation with the other applied method).
However, as the significance levels can still change with non-transformed data, and hence the relative importance of the findings can shift, it is necessary that the authors either argue why log 10 transformation is necessary to be kept, or they rerun the analyses.

Additional comments

This revised version has greatly improved in all aspects, and I’d like to thank the authors for taking my comments on the previous version seriously. The MS reads smoothly and the entire story, including figures, is much clearer and logical.

I have only a few extra comments in addition to the ones mentioned in the major 3 fields that should also be addressed for scientific completeness. The smaller issues are indicated in the annotated pdf.

1) In the discussion, regarding the bat humerus laminarity, you say: “…the bat humerus. The lack of laminarity in this (suspected) torsionally-loaded humerus is presumably due to the slow somatic growth rates of bats…”
This sounds a bit confusing to me, because above you related lower growth rates to laminarity, and here it sounds like laminarity was associated to fast growth, and hence absent in slow growing bats.
Furthermore, bat bone vascularity seems extremely low for a small mammal that otherwise shows high metabolic rates (when not in torpor of course) and, as far as I know, their growth is not outstandingly slow among mammals of their size. Hence, this pattern of low cortical vascularity (almost lizard-like) must be explained by other factors (which might well be due to that central artery that is not published unfortunately) than an extremely slow growth.
So I feel that arguing with the missing laminarity in bat bones is, in itself, misleading because the simple lack of or extremely low vascularity is not emphasized, although it’s very important in this context of discussion.
So it should be emphasized that bat bones are almost avascular in smaller species and have very sparse vascularization in larger species which makes bat bones hard to compare to the densely vascularized bird bones in general.

2) Concerning wing bones and their relation to LI, in my opinion, the fact that you detected that "ontogeny-independent growth rate" which explained a considerable amount of variance in wing elements also supports the release of canalizing selection pressures on the wing bones. This could also be discussed in the Discussion part as well as in the Conclusion, if you agree with my interpretation on this.

3) Please, also check your Tables in the annotated pdf where I indicated some missing data that need to be explained and some columns the meaning of which was clarified neither in the text, not in the table caption.

All in all, it is a very nice and useful study with very interesting findings.

Best wishes,
Edina Prondvai

---

## Round 0.3 · Minor Revisions

Thank you for the detailed revisions. I am convinced that they have addressed the reviewer's comments satisfactorily. However, I do not see the R script in the supp info or its location cited in the main text (apologies if somehow I missed it!). Please add a final sentence or two to the Methods stating where the R scripts are and all of the data required to re-run/modify all analyses. Once that data/methods availability is resolved, I can accept the MS. Thank you.

---

## Round 0.4 · accepted · Accept

Thank you! This helps a lot to have the data availability clearly laid out. Congrats on acceptance!